# Measuring the Level of Medical-Emergency-Related Knowledge among Senior Dental Students and Clinical Trainers

**DOI:** 10.3390/ijerph18136889

**Published:** 2021-06-27

**Authors:** Giath Gazal, Hamzah Aljohani, Khalid H Al-Samadani, Mohammad Zakaria Nassani

**Affiliations:** 1Department of Oral and Maxillofacial Surgery, Taibah University, Al Madinah Al Munawwarah 41311, Saudi Arabia; aljuhanihamzah@gmail.com; 2Department of Restorative Dental Science, College of Dentistry, Taibah University, Al Madinah Al Munawwarah 41311, Saudi Arabia; kalsamadani@gmail.com; 3Department of Restorative and Prosthetic Dental Sciences, College of Dentistry, Dar Al Uloom University, Riyadh 11512, Saudi Arabia; mznassani@hotmail.com

**Keywords:** knowledge, management, medical emergencies, dental chair

## Abstract

Objectives: This study aimed to measure the level of medical-emergency-related knowledge among senior dental students and clinical trainers in Saudi Arabia. Methods: This cross-sectional pilot survey was conducted at Taibah Dental College, Madina, Saudi Arabia between March 2017 and November 2018. Two hundred and seventy-five self-administered anonymous questionnaires on the management of common medical emergencies were distributed to all senior dental students and clinical trainers at Taibah Dental College. Results: There was a serious lack of knowledge regarding the management of medical emergency scenarios among the participants. Only 54% of participants knew the correct management for some frequent and life-threating conditions such as “crushed chest pain”, and only 30–35% of participants knew the correct management of deeply sedated patients with benzodiazepine overdose and crisis of hypoadrenalism. Moderate-quality knowledge (50–74% of participants responded correctly) was noted for the following conditions: sudden onset of brain stroke, psychiatric patient, unconscious patient with hypoglycemia, patient with postural hypotension, and patient with hyperventilation. Based on the scale of knowledge, there were significant differences in the level of knowledge between clinical trainers, senior dental students, and junior dental students (*p* ≤ 0.01). Almost all students and 90% of trainers declared the need for further training. Conclusions: The overall knowledge regarding the management of medical emergency crises in the dental chair was moderate. However, the scale of knowledge regarding the management of medical emergency crises has gradually increased with the number of years of experience. Most participants recognize the need for further training.

## 1. Advances in Knowledge

This study provides an informative platform for dental practitioners, enabling them to eliminate the uncertainty and confusion that they encounter when they are involved in a medical emergency case in the dental chair.

## 2. Application to Patient Care

This article aims to improve the quality of dental care provided to diabetic/heart disease patients in the dental office.

## 3. Introduction

Understanding emergency measures for life-threatening conditions that occur in the dental office enhance the self-confidence of dental practitioners and give them the ability to act properly in such circumstances [1]. Therefore, a great responsibility rests on the faculties of dentistry to provide an adequate curriculum and the necessary training in order to graduate dentists with the required level of understanding and training [2,3]. Knowledge of how to deal with patients who need dental treatments and who have life-threatening medical problems such as acute asthma, ischemic heart disease, uncontrolled diabetes, etc., enhance dentists’ ability to make clinical decisions with great confidence [1,2,3,4]. As a result, the patient feels that he/she is in expert, safe, and professional hands. Dentists have to recognize that medical emergencies can take place at any time in the daily dental practice. They should expect to encounter and manage at least one or two life-threatening medical emergencies over their clinical life [1]. Possession of adequate knowledge and competency to deal with medical emergencies will provide the dentist with greater confidence to manage one of the most critical aspects of his/her profession. The best approach to deal with an emergency situation is to adequately prepare yourself for such a critical moment [2]. Generally, an emergency plan should include training of the dental staff to manage emergencies, publication of emergency-related guidelines, and training on the immediate use of emergency kit or “crash cart” [3].

Despite the efforts made by dental schools to deliver the necessary knowledge about the management of medical emergency in the dental chair, there is still a number of dental graduates and dentists suffering from a deficit of knowledge in this regards [4]. This defect in knowledge was recognized by many studies; as an example, Jodalli and Ankola [4] reported a shallow level of knowledge in medical emergencies, related drugs, and equipment amongst dental graduates in Belgaum city of India. The participants in that study indicated that medical emergencies should be an essential topic of the dental study plan. Furthermore, most of the participating dental graduates expressed a will for more training in utilizing practical courses in medical emergencies [4]. Another study by de Bedout et al. [5] underlined apparent confusion among dental residents and practicing periodontists in handling of several medical emergency cases and varying degrees of incorrect diagnoses. It was also concluded that further training about case specific features is needed for the Indiana dental faculty members and residents [5,6,7,8]. In a survey among Saudi dental students and interns, the authors concluded that the availability of medications and basic emergency equipment in the dental settings coupled with well-trained dental staff on the management of basic life support have a major role in the reduction in mortality risk related to medical emergencies in the dental chair [3]. Additionally, Paquette et al. emphasized that dentists need to enhance their professional skills to positively contribute to the overall health management of their patients [8].

It is redundant to say that dentists should be effectively trained to handle medical emergencies. On the level of designing a dental curriculum, it is necessary to improve the quality of training in medical emergencies so as to enhance dental students’ ability to diagnose and deal with medical emergency cases in a professional manner [8,9]. While medical emergencies are wide in range, severe asthmatic attack, acute hypoglycemia, anaphylactic attack, and epilepsy attack are examples of the medical emergencies that dentists and students could encounter in the dental chair and require careful immediate management and adequate training to deal with.

In Saudi Arabia, it is a precondition for healthcare professionals to attend a training course and pass an assessment exam in cardiopulmonary resuscitation to obtain a work license [3]. Such a measure is a sign of good practice. It comes under the welfare of patients and clinicians, as it saves patients’ lives and protects health professionals from litigation. Despite the importance of being adequately qualified in the management of medical emergencies, little is known about the knowledge and experience on the management of common dental-office-related medical emergencies among dental students and trainers in Saudi Arabia.

The aim of this pilot survey was (a) to measure the level of medical-emergency-related knowledge and its relationship to the expertise and (b) to measure the awareness and the need for training on the management of medical emergencies among senior dental students and trainers at Taibah Dental College, Medina, Saudi Arabia.

## 4. Materials and Methods

This cross-sectional study was conducted at Taibah Dental College, Madina, Saudi Arabia between March 2017 and November 2018. The protocol was approved by the Ethics and Research Committee at Taibah Dental College (Study Reference No. TUCDREC-2l6l2l8-Juhani). The target population was 4th, 5th, internship dental students, and clinical trainers working as teaching staff members at Taibah Dental College. Lists of senior dental students and clinical trainers were obtained from the examination office at Taibah Dental College. This identified 275 subjects who were eligible to take part in this study (210 students and 65 clinical trainers). The study instrument was a self-administered questionnaire that was developed using the textbook of Master Dentistry (Guidance on the management of common medical emergencies; Coulthard et al. [6]). The questionnaire was presented in English. In the first stage, it was piloted among 15 dental students and 10 clinical trainers to ensure the clarity and feasibility of its contents.

In the first part of this questionnaire, participants were required to record their demographical information including age, gender, and nationality in addition to their years of experience. In the second part, participants were presented with 20 medical emergency cases that frequently take place in the dental chair [6] (Table 1).

Participants were asked to indicate on a 3-point scale their level of agreement with proposed management for each emergency situation. Scale categories were agree, disagree, I do not know. If the respondent agrees on the suggested management of the emergency case, that means a correct response. On the contrary, a disagreement indicates a wrong response. As well, “I do not know” indicates no/wrong response. A scale of knowledge was created by summing the number of correct answers.

In the last part of the questionnaire, each participant was required to self-evaluate his/her training, knowledge, and clinical competency in the management of medical emergencies.

Questionnaire envelopes were distributed to students by class leaders and by heads of departments to clinical trainers. Participants were ensured that participation was voluntary, anonymous, and the collected data would be confidential.

## 5. Statistical Analysis

Descriptive statistics were used to present characteristics of participants, and frequency tables were generated to illustrate the response of participants to questionnaire items; continuous data were expressed by mean ± standard deviation. Participants’ responses to questionnaire items were regrouped to either a correct or incorrect response; then, the scale of knowledge was calculated by summing the correct response (scale ranged from 0 to 20). Students were grouped according to their year in the faculty (4th year, 5th year, and internship), and trainers were grouped according to their years of clinical experience (<=10 years, between 11–20 years, >20 years). We used a Chi-square for trend to compare the responses for questionnaire items between the different groups of participants. The significance level was set at *p* ≤ 0.01. All statistical analyses were undertaken with SPSS software (IBM SPSS Statistics for Windows, Version 20.0, Released 2011, IBM Corp, Armonk, NY, USA).

## 6. Results

By the end of this survey, 202 out of 210 eligible students and 41 out of 65 eligible trainers completed and returned the study questionnaire. The final response rate was, therefore, 96% among dental students and 63% among clinical trainers. Characteristics of the study population are illustrated in Table 2.

All students were Saudi, compared to only 22% of trainers. The majority of dental students were 4th year students (70.8%). Trainers had a wide range of experience from less than 10 years (22%) to more than twenty years (34%). Many participants (students and trainers) showed a moderate level of knowledge in the management of medical emergencies, so for some frequent and life-threating conditions, such as “crushed chest pain”, only 54% responded correctly. The range of knowledge for this condition increased from 48% among 4th year students to 78% among trainers. The knowledge for the treatment of other conditions was even worse, so only 32% know the correct management of “deeply sedated patient with benzodiazepine overdose”. Details of responses for all conditions among study participants are presented in Table 3. Moderate-quality knowledge (50–74% of participants responded correctly) was present in the following conditions: crushed chest pain, sudden onset of brain stroke, psychiatric patient, unconscious patient with hypoglycemia, patient with postural hypotension, and patient with hyperventilation.

The scale of knowledge was, as expected, the lowest among the 4th year students (scale 12.7, 95% confidence interval 12.2–13.2), increasing to 13.6 (95% confidence interval 12.3–15) among the 5th year students. Among internship students and trainers with less than 20 years of experience, the scale was about 14.7. Among trainers with more than 20 years of experience, the scale increased to 16.2 (95% confidence interval: 14.6–17.9). Figure 1 shows the scale of knowledge for the different groups of participants.

Table 4 shows some variations among study groups in the self-evaluation of their training, knowledge, and clinical competency in the management of medical emergencies and the need for further training. It is clear that 61.7% of participants received training in the management of medical emergencies during an undergraduate dental program (range from 53.3% for 4th year dental students to 94.4% among trainers with 10–20 years of experience). Most students evaluated their training as fair, while most trainers evaluated their training as good. Satisfaction with the knowledge about medical emergencies was in general moderate with no specific difference between groups. The need for further training in the management of emergency cases was very high in general (97.9%). This need was declared by almost all students (only one student declares no need for further training), and even among trainers, more than 90% of them declared the need for further training. The details of these results are summarized in Table 4.

## 7. Discussion

Lack of knowledge and training on the management of dental chair medical emergency problems is considered a main source of anxiety affecting not only patient’s perception but also all junior and senior dental practitioners [7,8,9,10,11]. Therefore, a successful dental school is the one that has competent clinical trainers who contribute to students’ relevant professional development [8,9,10,11,12]. By establishing such an action, a competent dental student and trainer will be able to treat and safely handle any dental chair medical emergency condition with a minimal level of anxiety [7]. The findings of this pilot survey show a serious lack of knowledge regarding management of medical emergency scenarios among dental students and even among trainers. This deficit is reflected by high proportions of participants who have a lack of information or incorrect knowledge on managing these conditions. This lack is recognized by 97.9% of participants who declared the need for additional training. Such a lack must be addressed by medical education departments and educators in Saudi dental schools.

Although the overall knowledge was moderate, we noticed a higher knowledge rate for some conditions; so, 84% of participants knew the correct management of “sudden cardiac death”, and 75.3% for the management of “cardiopulmonary resuscitation steps”. This may be explained by the good training the students received from their trainers, attending and passing successfully the CPR course, which is considered as a prerequisite to obtaining a dental license in the KSA, and it is also a part of the dental curriculum. One of the most interesting findings from this study was the awareness and knowledge toward the management of a medical emergency in the dental chair has accumulated over time amongst the participants. The scale of knowledge, which ranges between 0 and 20, scored the lowest (12.7) among the 4th year students, increased to 13.6 among the 5th year students, then 14 among internship students, trainers with less than 20 years of experience was about 14.7, and the scale increased in trainers with greater than 20 years of experience to 16.2.

On the other hand, the results indicate that clinical trainers and dental students have a good level of knowledge and information regarding certain medical emergency scenarios, for example the treatment of conscious patients with hypoglycemia, epileptic seizures, acute asthmatic attacks, and chronic liver disease. This might be attributed to the fact that those conditions are widespread within this particular community [12,13,14,15,16,17,18]. Another interesting result was that students had better knowledge than their clinical trainers in terms of medical emergency management of conscious and unconscious patients with hypoglycemia, patients with renal failure who need dental extraction under local anesthesia, and psychiatric patients. This may be due to the current knowledge they have obtained from their curriculum.

A study by Dym et al. (2016) [11] indicated that medical emergencies may take place at any dental practice, and appropriate measures should be planned to enable the healthcare professionals to tackle such emergency cases at their practice place. High awareness of the dental team and their rapid reaction to any medical emergency situation can be regarded a crucial factor in the successful management of the emergencies occurring in the dental office [12]. However, most general dental practitioners lack the knowledge to deal with medical emergencies affecting pediatric patients and do not have the confidence to identify and handle emergencies in children [14,19,20]. Therefore, it was recommended to provide dentists with adequate training and education to overcome this professional defect [8]. Instant application of the right medical emergency protocol in the dental office can significantly improve the chance for a positive outcome and save the life of the affected patient [13]. In light of these findings, planners of dental education in Saudi Arabia should draw attention to the necessity of planning more lectures, practical sessions, and workshops to cover topics on the management of medical emergency problems for undergraduate dental students. This issue was resolved according to some other studies through the application of constructed sessions and workshops related to the management of medical emergency cases in the dental chair. For example, Miller and Metz [14] conducted a study on 120 dental students to evaluate the importance of clinical scenario videos in improving the perception of dental students of the basic sciences and their ability to apply content knowledge. Video modules were developed utilizing simulated patients and custom-designed animations to illustrate ways of management of medical emergencies in the dental practice. The study findings revealed that the video clips and animation helped dental students to understand and recognize the causes of medical emergencies and implement their understanding of physiology to the scenario and significantly improved their competency to answer various clinical questions. It can be argued that online modules may be a useful tool to promote students’ perceptions of the basic sciences and increase their ability to use basic science content to important scenarios in the clinical practice [21,22].

A similar study was carried out among graduating dental students at the University of British Columbia from 2008 until 2012 to analyze their perceptions regarding the importance of competency and confidence in their ability associated with each dental educational statement [15]. The results showed that, overall, students rated all the competencies as important, but they rated their confidence lower than the perceived importance. The competencies that were rated as most important by students were correlated with tasks regularly practiced throughout undergraduate dental education [15]. A survey study was carried out in Saudi Arabia between 2013 and 2014 to evaluate readiness for medical emergencies in private dental centers [22]. The outcome of this study revealed a deficiency in personnel training, drug availability, and emergency equipment among the investigated dental clinics. More rigorous rules and regulations for medical emergency education must be implemented to prevent disasters in these dental clinics [22]. Finally, knowledge about the management of medical emergencies in the dental chair is essential for dental graduates. A key of success in the field of dentistry depends on the recognition of the severity of medical conditions and taking the necessary action before and during dental treatment in order to avoid medical crises in the dental chair [23]. Yearly training can be of help to enhance students’ preparedness in the management of medical emergencies in the dental setting [24].

A limitation of this survey is related to the fact that the survey was conducted in one public university, which may affect the generalizability of the results to the whole dental students in KSA. First, second, and third year students did not participate in the study due to their lack of clinical experience. There was no participation of other local and international dental colleges because this survey is considered a pilot study. In order to confirm the findings of this study, a nationwide future survey is highly recommended.

## 8. Conclusions

The overall knowledge regarding the management of medical emergency crises in the dental chair was moderate. However, the scale of knowledge regarding the management of medical emergency crises has gradually increased with the number of years of experience. The need for further training in the management of emergency medical cases in the dental chair was very high in general. Online modules, clinical courses, and workshops on dental chair medical emergency management are recommended to improve clinical trainers’ and students’ basic medical knowledge and enhance their ability to use it in clinical practice.

## Figures and Tables

**Figure 1 ijerph-18-06889-f001:**
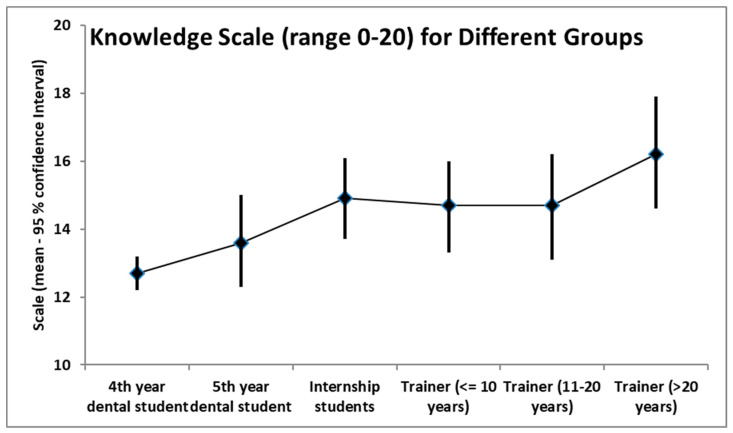
Knowledge scale for the different groups of participants. Knowledge scale ranges between 0 and 20.

**Table 1 ijerph-18-06889-t001:** Twenty medical emergency cases that frequently take place in the dental chair.

Medical Emergency Scenario	Proposed Management
1- First-line treatment of fainting patient in dental chair	Lay flat +/− give oxygen→expect prompt recovery
2- First-line treatment of patient with hyperventilation	Ask patient to re-breath from cupped hands or reservoir bag
3- First-line treatment of patient with postural hypotension in dental chair	Lay flat, give oxygen, and sit up very slowly
4- First-line treatment of conscious patient with hypoglycemia	Give glucose with a little water or glucose oral gel
5- First-line treatment of unconscious patient with hypoglycemia	1 mg Glucagon injection
6- First-line treatment of patient with epileptic seizure	Place the patient in recovery position and check airway after convulsive movements have subsided
7- First-line treatment of patient has a crisis of hypoadrenalism (Addison’s disease or long-term steroids) in dental chair	Lay flat + give oxygen + 200 mg hydrocortisone IV
8- First-line treatment of patient has acute asthmatic attack in dental chair	Sit up + give oxygen + salbutamol
9- First-line treatment of patient has anaphylactic shock	1 mL adrenaline (1 mg/mL) for IM
10- First-line treatment of patient has been deeply sedated with benzodiazepine overdose and suffered from severe respiratory depression in dental chair	Reverse of conscious sedation by giving 0.2 mg Flumazenil IV over 15 s
11- First-line treatment of patient has sudden onset of brain stroke (hemiplegia or quadriplegia) in dental chair	Reassure the patient and transfer to hospital
12- First-line treatment of psychiatric patient has unusual/bizarre/agitated/violent behaviour in dental chair	Transfer to hospital
13- First-line treatment of patient has crushed chest pain in dental chair	400 mcg nitroglycerin spray + 300 mg aspirin chewable tablets + 50% oxygen with 50% nitrous oxide
14- Further management of patient who has episode of chest pain relieved by rest and nitrates	Give oxygen and allow home if mild and rapidly recovered
15- Further management of patient who has severe chest pain (severe angina or myocardial infarction) not relieved by rest and nitrates	Hospital for giving: IV diamorphine + thrombolytic therapy (heparin)
16- First-line treatment of patient who has sudden heart arrest in dental chair	Immediate cardiopulmonary resuscitation (CPR)
17- Cardiopulmonary resuscitation (CRP) consists of:	After 30 chest compression, give 2 breaths (the 30:2 cycle of CPR)
18- First-line management of patient with chronic liver disease needs dental extraction under LA	Arrangement for a coagulation screen and liver function tests prior to surgery
19- First-line treatment of patient with known diabetes who becomes sweaty, with nausea and tachycardia in dental chair	Check if patient has eaten and give him some glucose
20- First-line management of patient with renal failure, needs dental extraction under LA	Dental extraction must be done the day following dialysis because there is no active heparin in circulation

**Table 2 ijerph-18-06889-t002:** Characteristics of participants.

Characteristics of Participants		Students (n = 202)	Trainers (n = 41)
Gender	Female	56.4% (114)	70.7% (29)
	Male	43.6% (88)	29.3% (12)
Nationality	Saudi	100% (202)	22% (9)
	Non-Saudi	0% (0)	78% (32)
Year of study (Students)	4th year	70.8% (143)	-
	5th year	17.3% (35)	-
	Internship	11.9% (24)	-
Experience (Trainers)	Trainer <= 10 years	-	22%(9)
	Trainer 11–20 years	-	43.9% (18)
	Trainer > 20 years	-	34.1% (14)
Qualification (Trainers)	Bachelor	-	2.4% (1)
	MSc	-	12.2% (5)
	Board	-	14.6% (6)
	PhD	-	70.7% (29)
		Mean ± SD	Mean (SD)
Age		22.9 ± 1.3	43.8 ± 8.2
Clinical experience in years (Trainers)		-	18.8 ± 8.7

**Table 3 ijerph-18-06889-t003:** Percentages of correct answers of participating dental students and clinical trainers to the management of 20 medical emergency cases that frequently take place in the dental chair.

Medical Emergency Scenario	4th Year (N = 143) % (95% CI)	5th Year (N = 35) % (95% CI)	Internship (24) % (95% CI)	Trainer (41) % (95% CI)	*p*
Fainting patient in dental chair	72.7% (64.7–79.8%)	60% (42.1–76.1%)	83.3% (62.6–95.3%)	92.7% (80.1–98.5%)	0.01 *
Patient with hyperventilation	65.7% (57.3–73.5%)	62.9% (44.9–78.5%)	91.7% (73–99%)	70.7% (54.5–83.9%)	0.17
Patient with postural hypotension	65.7% (57.3–73.5%)	74.3% (56.7–87.5%)	79.2% (57.8–92.9%)	90.2% (76.9–97.3%)	0.001 *
Conscious patient with hypoglycemia	96.5% (92–98.9%)	100% (90–100%)	100% (85.8–100%)	92.7% (80.1–98.5%)	0.49
Unconscious patient with hypoglycemia	75.5% (67.6–82.3%)	65.7% (47.8–80.9%)	75% (53.3–90.2%)	63.4% (46.9–77.9%)	0.16
Epileptic seizure	73.4% (65.4–80.5%)	80% (63.1–91.6%)	87.5% (67.6–97.3%)	85.4% (70.8–94.4%)	0.05 *
Crisis of hypoadrenalism	30.1% (22.7–38.3%)	45.7% (28.8–63.4%)	50% (29.1–70.9%)	61% (44.5–75.8%)	<0.001 *
Acute asthmatic attack	68.5% (60.2–76%)	85.7% (69.7–95.2%)	91.7% (73–99%)	90.2% (76.9–97.3%)	0.001 *
Anaphylactic shock	50.3% (41.9–58.8%)	71.4% (53.7–85.4%)	79.2% (57.8–92.9%)	70.7% (54.5–83.9%)	0.002 *
Deeply sedated with benzodiazepine overdose	24.5% (17.7–32.4%)	54.3% (36.6–71.2%)	41.7% (22.1–63.4%)	34.1% (20.1–50.6%)	0.08
Sudden onset of brain stroke	67.1% (58.8–74.8%)	68.6% (50.7–83.1%)	70.8% (48.9–87.4%)	68.3% (51.9–81.9%)	0.8
Psychiatric patient	60.8% (52.3–68.9%)	65.7% (47.8–80.9%)	62.5% (40.6–81.2%)	56.1% (39.7–71.5%)	0.69
Crushed chest pain	47.6% (39.1–56.1%)	37.1% (21.5–55.1%)	70.8% (48.9–87.4%)	78% (62.4–89.4%)	<0.001 *
Episode of chest pain relieved by rest and nitrates	50.3% (41.9–58.8%)	51.4% (34–68.6%)	50% (29.1–70.9%)	73.2% (57.1–85.8%)	0.03 *
Severe chest pain not relieved by rest and nitrates	51.7% (43.2–60.2%)	37.1% (21.5–55.1%)	54.2% (32.8–74.4%)	68.3% (51.9–81.9%)	0.11
Sudden heart arrest	81.8% (74.5–87.8%)	74.3% (56.7–87.5%)	87.5% (67.6–97.3%)	97.6% (87.1–99.9%)	0.02 *
Cardiopulmonary resuscitation steps	71.3% (63.2–78.6%)	82.9% (66.4–93.4%)	70.8% (48.9–87.4%)	85.4% (70.8–94.4%)	0.1
Chronic liver disease needs dental extraction	69.2% (61–76.7%)	74.3% (56.7–87.5%)	91.7% (73–99%)	85.4% (70.8–94.4%)	0.008
Known diabetes who becomes sweaty, with nausea and tachycardia	73.4% (65.4–80.5%)	91.4% (76.9–98.2%)	75% (53.3–90.2%)	85.4% (70.8–94.4%)	0.11
Renal failure needs dental extraction	71.3% (63.2–78.6%)	80% (63.1–91.6%)	79.2% (57.8–92.9%)	70.7% (54.5–83.9%)	0.79

* Significant difference in percentages of correct answers of participating dental students and clinical trainers to the management of 20 medical emergency cases that frequently take place in the dental chair (*p* < 0.001, Student’s t-test).

**Table 4 ijerph-18-06889-t004:** Participants’ self-evaluation of their training, knowledge, and clinical competency in the management of medical emergencies.

	4th Year Dental Student (143)	5th Year Dental Student (35)	Internship Dental Student (24)	Trainer <=10 Years (9)	Trainer 11–20 Years (18)	Trainer >20 Years (14)	*p*
Received training in the management of medical emergencies during Undergraduate dental program	53.8% (77)	71.4% (25)	58.3% (14)	88.9% (8)	94.4% (17)	64.3% (9)	0.004 *
Quality of training	-	-	-	-	-	-	0.07
Poor	15.6% (12)	4% (1)	14.3% (2)	0% (0)	11.8% (2)	0% (0)	
Fair	33.8% (26)	52% (13)	50% (7)	37.5% (3)	11.8% (2)	22.2% (2)	
Good	37.7% (29)	36% (9)	35.7% (5)	50% (4)	41.2% (7)	77.8% (7)	
Very good	9.1% (7)	8% (2)	0% (0)	12.5% (1)	29.4% (5)	0% (0)	
Excellent	3.9% (3)	0% (0)	0% (0)	0% (0)	5.9% (1)	0% (0)	
Satisfaction level with knowledge about medical emergencies	-	-	-	-	-	-	-
Not at all satisfied	20.3% (29)	14.3% (5)	25% (6)	11.1% (1)	5.6% (1)	0% (0)	<0.001 *
Slightly satisfied	40.6% (58)	42.9% (15)	45.8% (11)	11.1% (1)	5.6% (1)	7.1% (1)	
Moderately satisfied	30.8% (44)	42.9% (15)	25% (6)	55.6% (5)	50% (9)	64.3% (9)	
Very satisfied	7% (10)	0% (0)	4.2% (1)	22.2% (2)	27.8% (5)	28.6% (4)	
Extremely satisfied	1.4% (2)	0% (0)	0% (0)	0% (0)	11.1% (2)	0% (0)	
Self-evaluation of the clinical competency in dealing with medical emergencies	-	-	-	-	-	-	<0.001 *
Poor	18.9% (27)	22.9% (8)	25% (6)	11.1% (1)	5.6% (1)	0% (0)	
Fair	44.1% (63)	45.7% (16)	50% (12)	33.3% (3)	0% (0)	35.7% (5)	
Good	31.5% (45)	28.6% (10)	25% (6)	33.3% (3)	55.6% (10)	35.7% (5)	
Very good	4.9% (7)	2.9% (1)	0% (0)	22.2% (2)	22.2% (4)	21.4% (3)	
Excellent	0.7% (1)	0% (0)	0% (0)	0% (0)	16.7% (3)	7.1% (1)	
Need for further training in the field of medical emergencies	99.3% (142)	100% (35)	100% (24)	88.9% (8)	88.9% (16)	92.9% (13)	0.002 *

* denotes significant difference at *p* ≤ 0.01, *p* was calculated according to Chi-square for trend.

## Data Availability

The data that support findings of this study are available on request from the corresponding author.

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
