# Peer review of "Measuring the Level of Medical-Emergency-Related Knowledge among Senior Dental Students and Clinical Trainers"

_ijerph, 2021, doi:10.3390/ijerph18136889_

Round 1

Reviewer 1 Report

Pilot: is this paper a pilot study?

You say: “This cross-sectional pilot survey was conducted at …between March 2017 and November 2018. Two-hundred and seventy-five self-administered anonymous questionnaires.  But also, In the first stage, it was piloted among 15 dental students and 10 clinical trainers to ensure the clarity and  feasibility of its contents”

Junior dental students are they included in the survey?

The target population wear clinical trainers working as teaching staff members and dental students, only senior students or senior and junior dental  students?

You say: “Based on the scale of knowledge, there were significant differences in the level of knowledge between clinical trainers, senior dental and junior dental students"

You can conclude that the need for further training in the management of emergency cases was very high in general, but I think you can not conclude: “ Online modules, clinical courses, and workshops on dental chair medical emergency management might improve clinical trainers’ and students' basic medical knowledge and enhance their ability to use it in clinic “ that is your opinion not the result of the survey

You should checked the test, there are some spelling mistakes

Author Response

Reviewer 1 comments:

  1. Pilot: is this paper a pilot study? Answer: yes this is a pilot survey because we are planning to conduct a larger study that includes all dental colleges in the Kingdom, in addition to neighboring countries in the near future. However, we did a pilot survey before the launch of the main study to validate our questionnaire and ensure the clarity and feasibility of its contents (15 dental students and 10 clinical trainers).
  2. Junior dental students are they included in the survey? Answer: No junior dental student were not included in this survey because dental students in the first three years of study, their knowledge is still superficial, in addition to the lack of comprehensive clinical treatments.
  3. You can conclude that the need for further training in the management of emergency cases was very high in general, but I think you cannot conclude: “ Online modules, clinical courses, and workshops on dental chair medical emergency management might improve clinical trainers’ and students' basic medical knowledge and enhance their ability to use it in clinic “ that is your opinion not the result of the surveyAnswer: the following sentence was added it to conclusion: However, the rest of conclusion was converted to recommendation.See line 289
  4. “The need for further training in the management of emergency medical cases on the dental hair was very high in general.” See line 287-28
  5. You should checked the test, there are some spelling mistakes Answer: Spelling mistakes were corrected……Done.

Reviewer 2 Report

I am grateful for the opportunity to review the manuscript entitled "Measuring the level of medical emergency-related knowledge among senior dental students and clinical trainers".

Authors are required to make a few modifications in order to consider their manuscript for publication:

1) The authors' affiliation must be added.

2) Why are results obtained in 2017 and 2018 published in 2021?

3) Authors should make a careful revision of the grammar of the manuscript.

4) Why were only fourth and fifth year students considered?

5) Has this questionnaire been used before in similar studies?

6) Have the authors not considered doing this survey in other universities to increase the number of participants?

7) Have the characteristics of the participants taken into account their previous education?

8) The authors should describe any limitations of their study.

9) Has this study been evaluated by a Bioethics Committee?

10) Bibliographical references are not described according to the journal's guidelines.

Author Response

Reviewer 2 comments:

1) The authors' affiliation must be added.

Answer: all affiliations were added into the authors section during the submission process.

Giath Gazal

Department of Oral and Maxillofacial Surgery, Taibah University, Al Madinah Al Munawwarah, Saudi Arabia

Hamzah Aljohani

Department of Oral and Maxillofacial Surgery, Taibah University, Al Madinah Al Munawwarah, Saudi Arabia

Khalid H Al-Samadani

Department of Restorative Dental Science, College of Dentistry, Taibah University, Al Madinah Al Munawwarah, Saudi Arabia

Mohammad Zakaria Nassani

Department of Restorative and Prosthetic Dental Sciences, College of Dentistry, Dar Al Uloom University, Riyadh, Saudi Arabia.

2) Why are results obtained in 2017 and 2018 published in 2021?

Answer: Collection of data was done in the 2018, and statistical analysis was carried out in 2019. Our target for publication was 2020. However, unfortunately Covid-19 pandemic hit the world and there were lockdown everywhere in the country. Moreover, the main author Dr. Gazal and his families were infected by the Covid-19 and he was on sick leave for 6 months.

3) Authors should make a careful revision of the grammar of the manuscript.

Answer: Done

4) Why were only fourth and fifth year students considered?

Answer: Because dental students in the first three years of study, their knowledge is still superficial, in addition to the lack of comprehensive clinical treatments. In our college dental students start giving local anesthetic injections and carry on teeth extractions in their fourth and fifth year of study.

5) Has this questionnaire been used before in similar studies?

Answer: No it was not. This questionnaire was developed using the textbook of Master Dentistry (Guidance on the management of common medical emergencies; Coulthard et al). Line 102-102

6) Have the authors not considered doing this survey in other universities to increase the number of participants?

Answer: This is a pilot survey because we are planning to conduct a larger study that includes all dental colleges in the Kingdom, in addition to neighboring countries in the near future.

7) Have the characteristics of the participants taken into account their previous education?

Answer: Yes because the questionnaire items were built in a way that requires the participant to use his/her cumulative experience in order to be able to answer the presented scenarios.

8) The authors should describe any limitations of their study.

Answer: done please see lines 277 -280

A limitation of this survey is related to the fact that the survey was conducted in one public university which may affect the generalizability of results to the whole dental students in KSA. In order to confirm the

findings of this study a nationwide future survey is highly recommended

9) Has this study been evaluated by a Bioethics Committee?

Answer: The protocol was approved by the Ethics and Research Committee at Taibah Dental College. Lines: 96-97

10) Bibliographical references are not described according to the journal's guidelines.

Answer: Done

Reviewer 3 Report

Dear Authors,

Congratulations on the successful completion and submission of your research manuscript for the peer-review process.

The present study highlights the current level of knowledge and awareness on management of medical emergencies that can be potentially encountered by dental students/clinicians in their dental practice. The study was conducted in a Dental College in Saudi Arabia that involved dental students and clinical trainers as the study participants who were asked to fill a questionnaire as part of the study. The results presented the scale of knowledge across the different groups of the participants and also highlighted their indication for the need for further training in the same. The authors concluded that the overall knowledge on the management of medical crisis in a dental setting was moderate with almost all of the participants expressing their interest in additional training. 

The strengths of the study include well-presented statistical data, inclusion of diverse medical emergency scenarios in the questionnaire, well-written discussion, significant emphasis on training the dental students to manage medical emergencies efficiently during their dental practice that can in turn have a great impact on their confidence and quality of dental care. 

Kindly consider the following suggestions:

Page 1, Line 28,29: Kindly rephrase this sentence for grammatical correction.

Page 1, Line 32: Kindly rewrite this sentence to make it easily absorbable, can change it to something like this : 'This article aims to improve the quality of dental care rendered to diabetic/heart disease patients in the dental office'.

Page 1, Line 37,38: can change to '...emergency measures for life-threatening conditions that occur in the dental office enhance ...'

Page 1, Line 40: change to '... to provide an adequate..'

Page 2, Line 73: 'professional skills'

Page 2, Line 82: can change to something like this '...emergencies that dentists and students could encounter on the dental chair and require..'

Page 2, Line 93: 'its'

Page 4, Line 119: Kindly indicate what happens when the participant  chooses 'I do not know'

Thank You!

Author Response

Reviewer 3 comments:

Page 1, Line 28,29: Kindly rephrase this sentence for grammatical correction.

Answer: done

This study provides an informative platform for dental practitioners, enabling them to eliminate the uncertainty and confusion that they encounter when they are involved in a medical emergency case on the dental chair.

Page 1, Line 32: Kindly rewrite this sentence to make it easily absorbable, can change it to something like this: 'This article aims to improve the quality of dental care rendered to diabetic/heart disease patients in the dental office'.

Answer: done see line 33 page 1

Page 1, Line 37,38: can change to '...emergency measures for life-threatening conditions that occur in the dental office enhance ...'

Answer: done. See page 1 line 37-38

Page 1, Line 40: change to '... to provide an adequate..'

Done

Page 2, Line 73: 'professional skills'

Done

Page 2, Line 82: can change to something like this '...emergencies that dentists and students could encounter on the dental chair and require..'

Done

Page 2, Line 93: 'its'

Done

Page 4, Line 119: Kindly indicate what happens when the participant chooses 'I do not know'

Done - As well, “I do not know” indicates no/wrong response.. Line 119

Round 2

Reviewer 2 Report

On the first page of the manuscript authors should indicate their affiliation.

In the limitations of the study the authors should indicate that they did not consider first, second and third year students.

Authors should expand on the limitations of their study.

What is the reference of the Bioethics Committee? This reference should be included in the manuscript.

Author Response

Reviewer comments:

  1. On the first page of the manuscript authors should indicate their affiliation.    Department of Restorative and Prosthetic Dental Sciences, College of Dentistry, Dar Al Uloom University, Riyadh, Saudi Arabia.Department of Restorative Dental Science, College of Dentistry, Taibah University, Al Madinah Al Munawwarah, Saudi ArabiaDepartment of Oral and Maxillofacial Surgery, Taibah University, Al Madinah Al Munawwarah, Saudi Arabia. Done please see page 1
  2. In the limitations of the study the authors should indicate that they did not consider first, second and third year students. Authors should expand on the limitations of their study.

A limitation of this survey is related to the fact that the survey was conducted in one public university which may affect the generalizability of the results to the whole dental students in KSA. First, second and third year students did not participate in the study due to their lack of clinical experience. Non-participation of other local and international dental colleges because this survey is considered a pilot study. In order to confirm the findings of this study a nationwide future survey is highly recommended.

3. What is the reference of the Bioethics Committee? This reference should be included in the manuscript.Done: The protocol was approved by the Ethics and Research Committee at Taibah Dental College (Study Reference No. TUCDREC-2l6l2l8-Juhani). See page 3 (103-104)  
